# Review of systematic reviews of non-pharmacological interventions to improve quality of life in cancer survivors

Morvwen Duncan,[1] Elisavet Moschopoulou,[2] Eldrid Herrington,[3,4] Jennifer Deane,[1] Rebecca Roylance,[5] Louise Jones,[6] Liam Bourke,[7,8] Adrienne Morgan,[9] Trudie Chalder,[10] Mohamed A Thaha,[3,4] Stephanie C. Taylor,[11] Ania Korszun,[12] Peter D. White,[12] Kamaldeep Bhui,[12] on behalf of SURECAN Investigators

MD and EM are joint first authors.

For numbered affiliations see end of article.

**Correspondence to**
Professor Kamaldeep Bhui;
k.s.bhui@qmul.ac.uk

## ABSTRACT

**Objectives** Over two million people in the UK are living with and beyond cancer. A third report diminished quality of life.

**Design** A review of published systematic reviews to identify effective non-pharmacological interventions to improve the quality of life of cancer survivors.

**Data sources** Databases searched until May 2017 included PubMed, Cochrane Central, EMBASE, MEDLINE, Web of Science, the Cumulative Index to Nursing and Allied Health Literature, and PsycINFO.

**Study selection** Published systematic reviews of randomised trials of non-pharmacological interventions for people living with and beyond cancer were included; included reviews targeted patients aged over 18. All participants had already received a cancer diagnosis. Interventions located in any healthcare setting, home or online were included. Reviews of alternative therapies or those non-English reports were excluded. Two researchers independently assessed titles, abstracts and the full text of papers, and independently extracted the data.

**Outcomes** The primary outcome of interest was any measure of global (overall) quality of life.

**Analytical methods** Quality assessment assessing methdological quality of systematic reviews (AMSTAR) and narrative synthesis, evaluating effectiveness of non-pharmacological interventions and their components.

**Results** Of 14430 unique titles, 21 were included in the review of reviews. There was little overlap in the primary papers across these reviews. Thirteen reviews covered mixed tumour groups, seven focused on breast cancer and one focused on prostate cancer. Face-to-face interventions were often combined with online, telephone and paper-based reading materials. Interventions included physical, psychological or behavioural, multidimensional rehabilitation and online approaches. Yoga specifically, physical exercise more generally, cognitive behavioural therapy (CBT) and mindfulness-based stress reduction (MBSR) programmes showed benefit in terms of quality of life.

**Conclusions** Exercise-based interventions were effective in the short (less than 3–8 months) and long term. CBT and MBSR also showed benefits, especially in the short

### Strengths and limitations of this study

► This is a systematic review of reviews and evidence synthesis of non-pharmacological interventions in cancer survivors.

► Longer term studies are needed and studies of greater methodological quality that adopt similar reporting standards.

► Definitions of survivor varied and more studies are needed for different types of cancer, and specifically for patients who have poor quality of life.

► More studies are needed that investigate educational, online and multidisciplinary team-based interventions.

► This review has some limitations in the methodology. Studies not in English and grey literature were not included. This was a review of reviews: we did not review individual studies focused on specific cancers or stage, and we did not reassess the quality of the primary studies included in each review.

term. The evidence for multidisciplinary, online and educational interventions was equivocal.

## INTRODUCTION

Advances in public awareness, early detection and improved treatments mean that more people are now living with and beyond cancer. For example, Cancer Research UK reports that 50% of people diagnosed with cancer in England and Wales survive 10 years or more, and survival rates have doubled over the last 40 years.[1] This group of survivors includes people at various stages of active treatment, and those in remission, who are gradually restoring their social and occupational roles.

A significant proportion of cancer survivors experience poor quality of life (QoL).[2] The main causes of poor QoL include depression,

anxiety, distress, fear of recurrence and lower levels of social support; impacts on relationships, family and social function; and psychological and social needs, and problems coping.[2] [3] The process of diagnosis and treatment is traumatic and disruptive. It is not unusual for patients with cancer to experience distress. Common experiences for those living with and beyond cancer include reduced physical ability, fatigue, changes in sexual activity and developing other medical conditions that affect function for many years.[2] [3] If a person is suffering from fatigue, depression or anxiety, they are understandably less motivated to visit friends or engage in social activities; the strain on marital relationships may lead to a loss of support: 25% of people who experience difficulties have broken up with their partner as a result of cancer.[3] [4] Thus, the effects of cancer extend beyond the diagnostic and active treatment phases. This review aims to gather the evidence for practitioners, patients and their carers about effective non-pharmacological interventions to improve QoL in cancer survivors. We sought to summarise the effectiveness of non-pharmacological interventions in cancer survivors as part of an (National Institute of Health Research) NIHR-funded programme development grant to inform the design and delivery of a full programme grant.

## METHODS
This review of reviews examined existing systematic reviews of non-pharmacological interventions that include information on QoL of those living with and beyond cancer.

### Inclusion and exclusion criteria
The study included any systematic reviews that explicitly reported randomised controlled trials (RCTs). Inclusion criteria were organised in accordance with the patient, intervention, comparison, outcome (PICO) reporting structure (see table 1). The population of interest was people living with and beyond cancer, who were aged 18 years or more, and who had received their cancer diagnosis as adults.

We defined non-pharmacological interventions as those that did not involve any drug or medicine, but they could include educational, behavioural, psychosocial approaches or physical activity; we excluded complementary and alternative therapies as defined by the NHS Choices resource.[5] However, we included physical activity and psychological approaches that were part of yoga-based interventions after consulting with patients in the development of the review. Comparators were not specified for the purpose of the inclusion criteria of the review of reviews, but comparators reported in the original reviews were considered in the analysis.

The primary outcome was QoL defined by physical, psychological and social functioning. We reported on studies that used an established and validated measure of global or overall QoL; some of these are cancer-specific. In the literature, the terms 'Quality of Life' and 'Health Related QoL' are used interchangeably; therefore, both are included under the term 'QoL' in this review. The study settings included any healthcare venue, such as hospital inpatient or outpatient services and community services, and also included home and remote e-technology-based interventions.

### Data sources
We searched the following databases: PubMed, Cochrane Central, EMBASE, MEDLINE, Web of Science, the Cumulative Index to Nursing and Allied Health Literature, and PsycINFO. The final search was from inception to May 2017 and is shown in online supplementary annex 1. We consulted experts in the field to assess completeness of the list of identified reviews, and where necessary contacted authors to secure the full-text versions.

### Study selection
Two authors (MD, JD) independently screened all titles and abstracts of studies identified by the search strategy against inclusion and exclusion criteria, and when eligibility was determined the full text was read. Discrepancies around inclusion were resolved by discussion or in consultation with

| Table 1 | Application of the PICO search strategy |
|---|---|
| Population | Participants living beyond cancer, who have completed active treatment with curative intent, aged 18 or more who received their cancer diagnosis in adulthood |
| Intervention | Non-pharmacological interventions: psychological, social and physical activity, excluding complementary and alternative therapies or medicines, including yoga interventions with meditation, activity or mindfulness |
| Outcomes | Quality of life |
| Setting | Any healthcare setting: hospital (inpatient or outpatient), community or remote (eg, using e-technology) |
| Study design | Systematic reviews that had explicitly searched for randomised controlled trials (RCTs); to be classified as a systematic review if the following criteria were met:<br>► clear inclusion criteria<br>► a systematic search strategy<br>► a screening procedure to identity relevant studies<br>► systematic data extraction and analysis procedures for RCTs |

PICO, Population, Intervention, Comparison, Outcome.

a third author when required (KB). We searched the reference lists of all included reviews to identify any further relevant reviews. The research team was not blinded to authors. Citations were downloaded and managed in an EndNote library.

## Data extraction

Two authors (EM, EH) independently extracted data from each of the eligible reviews into a purpose-built, predesigned, structured template. The data extraction forms were then summarised in a table and reviewed independently by a third reviewer (KB). Extracted data included the following information:

► Publication details: author, year, title, journal, country and format of publication.
► Study characteristics: number of primary studies, total number of participants, range of publication dates, gender, age range of participants and socioeconomic data, primary cancer site, length of time since final cancer treatment, and type of treatment.
► Intervention design and evaluation: setting, description of the intervention and its components: physical components, psychosocial components, educational components; duration of intervention, follow-up, number of treatment contacts, type of practitioner providing treatment, mode of delivery of intervention, and any outcomes.
► Documents: availability of treatment manuals.
► Results: main outcome measures, secondary outcome measures, narrative findings, adherence levels, patient satisfaction and effect sizes against intervention components.

## Assessment of methodological quality of included reviews

The methodological quality of the systematic reviews was evaluated using Assessing Methdological Quality of Systematic Reviews (AMSTAR),[6] a measurement tool for the assessment of multiple systematic reviews that has good reliability and validity (table 2). The AMSTAR checklist used can be found at https://amstar.ca/Amstar_Checklist.php.

## Data analysis and narrative synthesis

The intervention components were listed, followed by a narrative synthesis.[7] This included understanding components of the interventions, exploring patterns of findings across studies and within primary reviews, and giving greater weight to studies of higher quality in the interpretation of the findings, especially if there were contradictions between the findings of reviews. Ultimately, the purpose was to put into text format the key findings from the most robust evidence available, to guide treatment and future research recommendations. The synthesis set out reported effect sizes across studies, means and SD. Meta-analysis was not undertaken, due to heterogeneity of methods, outcomes and absence of reported effect sizes (10 reviews did not provide effect sizes). The publications were segmented into those reporting meta-analyses to

**Table 2** Assessing Methdological Quality of Systematic Reviews (AMSTAR), tool for the assessment of multiple systematic reviews

| Review | AMSTAR score* | Quality rating |
|---|---|---|
| Bourke et al[28] | 3 | Low |
| Buffart et al[11] | 6 | Moderate |
| Cramer et al[23] | 9 | High |
| Culos-Reed et al[14] | 3 | Low |
| Duijts et al[9] | 4 | Moderate |
| Ferrer et al[19] | 8 | High |
| Fong et al[10] | 8 | High |
| Fors et al[24] | 5 | Moderate |
| Galvão and Newton[13] | 2 | Low |
| Gerritsen and Vincent[20] | 6 | Moderate |
| Huang et al[27] | 8 | High |
| Khan et al[8] | 10 | High |
| McAlpine et al[15] | 5 | Moderate |
| Mewes et al[18] | 5 | Moderate |
| Mishra et al[12] | 10 | High |
| Osborn et al[17] | 7 | Moderate |
| Smits et al[21] | 8 | High |
| Spark et al[25] | 6 | Moderate |
| Spence et al[16] | 5 | Moderate |
| Zachariae and O'Toole[22] | 5 | Moderate |
| Zeng et al[26] | 6 | Moderate |

*The maximum score on AMSTAR is 11 and scores of 0–3 indicate that the review is of low quality, 4–7 of moderate quality and of 8–11 as high quality.

which the greatest weighting was given in the synthesis; some reviews did not undertake or report meta-analyses but rather reported each study, trends and the range of effect sizes; a third group reported no effect sizes but provided narrative statements.

## Patient and public involvement

Patients and carers (and respective organisations) were involved in the design and development of the programme development grant application (from which this review is one output). Patients and carers attended all the steering group meetings and were an integral part of the research team, commenting on and critiquing the inclusion and exclusion criteria, outcome selection, and the acceptability and likely value of interventions. As part of the steering group, they received and commented on study progression, emergent findings and reports. They are integral to the dissemination plans, including sharing the publication, but also helping craft lay summaries of the overall research project and key findings. A public-patient representative (EH) performed the data extraction together with research and clinical colleagues,

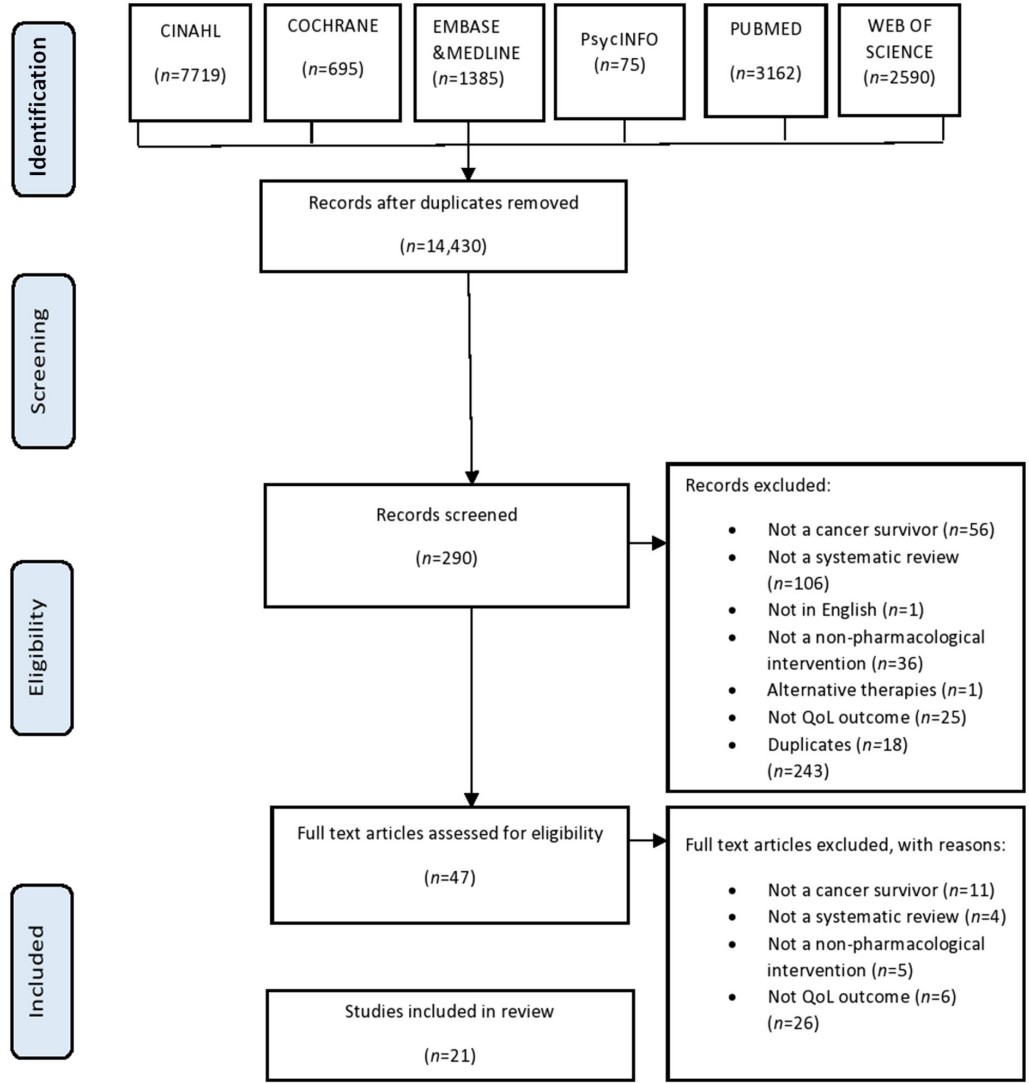

**Figure 1** Preferred Reporting Items for Systematic Reviews and Meta-Analyses (PRISMA) flow diagram of study selection. CINAHL, Cumulative Index to Nursing and Allied Health Literature; QoL, quality of life.

and coauthored and edited the review. Public-patient representatives were also part of the steering group and informed the design and delivery of the review.

## RESULTS
### Study selection
Electronic database searches yielded 14 430 unique reviews. From this 290 were included from the title search, followed by 47 from the abstract search. After scrutinising the full texts, 21 of eligible published reviews were included in this review (figure 1). The 26 excluded studies are listed in an online supplementary file. The quality scores are shown in table 2.

### Study characteristics
The types of interventions, settings, cancer type, measures of QoL and the key narrative findings are reported in table 3.

### Participants
The number of patients included in the reviews ranged from 262[8] to 7164.[9] Thirteen reviews covered mixed tumour groups,[10–22] seven specifically focused on breast cancer[8 9 23–27] and one on prostate cancer.[28]

### Intervention type and components
Face-to-face delivery of interventions was often combined with online delivery (three reviews)[9 24 28]; others included telephone communication (five reviews)[9 11 23 25 26] and printed information (two reviews).[11 25] Four reviews included interventions that provided supplementary compact discs, manuals or video tools.[11 23 24 28] Two reviews were from inpatient rehabilitation.[8 18] None of the reviews reported the use of structured manuals, and interventions were often not fully described or broken down into different components, nor was there attention to a mechanism or theory of change.

Ten of the reviews focused on physical interventions,[10 12 13 16 19–21 25 26 28] and three focused on

**Table 3** Characteristics of included reviews

| Review | Aims of review | Primary studies (n) | Participants | Definition of 'survivor' | Setting | Intervention, duration and frequency | Outcome—QoL measures | Narrative findings |
|---|---|---|---|---|---|---|---|---|
| Buffart et al[11] | Systematic review of RCTs and meta-analysis of the effects of yoga in cancer patients and survivors | 16 publications/13 RCTs | 744 patients with breast cancer and 39 patients with lymphoma during and after treatment Mean age range: 44–63 years | Patients during and after treatment | Face to face, with supplementary booklets and audiotapes of exercises for home practice | All included a yoga programme led by experienced yoga instructors with physical poses (asanas), breathing techniques, (pranayama), and relaxation or meditation (savasana or dhanya) Programme duration: 6 weeks to 6 months | FACT-G, SF-36, EORTC QLQ-C30, FLIC | Yoga has strong beneficial effects on distress, anxiety and depression, moderate effects on fatigue, general HRQoL, emotional function and social function, small effects on functional well-being, and no significant effects on physical function and sleep disturbances. |
| Bourke et al[28] | To evaluate the evidence from RCTs of supportive interventions designed to improve prostate cancer-specific QoL | 20 RCTs | 2654 prostate cancer survivors | Patients during and after treatment | Group or face to face, online or with supplementary audiotapes | Lifestyle interventions including exercise interventions, diet interventions or a combination of exercise and diet Multidisciplinary group education or online education and support Enhanced standard care interventions and cognitive behavioural interventions Varied durations and follow-up frequencies | FACT-P, QLQ-PR25, EPIC, EPIC-26, UCLA-PCI, PCa-QoL | Supervised and individually tailored patient-centred interventions such as lifestyle programmes are beneficial. |
| Cramer et al[23] | To systematically assess and meta-analyse the evidence for the effects of yoga on HRQoL and psychological health in patients with breast cancer and survivors | 12 RCTs were included in the qualitative synthesis and 10 of them were included in the meta-analysis. | 742 patients with breast cancer during or after treatment Mean age range: 44–63 years | Those who had completed active treatment before the onset of the study | Face to face, with supplementary audio and video tools or telephone calls | Yoga interventions including Iyengar yoga, Yoga of Awareness, Viniyoga, restorative yoga, yoga based on Patanjali's yoga tradition, Yoga in Daily Life, integrated yoga and hatha yoga Duration: 1 week to 6 months Frequency varied from daily sessions to weekly. | FACT-G, FACT-B, FACIT-Sp, SF-36, SF-12, FLIC, EORTC QLQ-C30 | There is moderate evidence for the short-term effect of yoga on global HRQoL. However these short-term effects could not be clearly distinguished from bias. |

Continued

**Table 3** Continued

| Review | Aims of review | Primary studies (n) | Participants | Definition of 'survivor' | Setting | Intervention, duration and frequency | Outcome—QoL measures | Narrative findings |
|---|---|---|---|---|---|---|---|---|
| Culos-Reed et al[14] | To determine the clinical significance of patient-reported outcomes from yoga interventions conducted with cancer survivors | 13 studies/7 RCTs | 474 patients with mixed cancer The majority were patients with breast cancer during and after treatment. RCTs: sample size in the treatment group at time 2 ranged from 13 to 45 patients. Mean age range: 46–60 years | Patients both on and off treatment | Face to face | Yoga styles included hatha, integral, Iyengar, Tibetan, Viniyoga and Vivekananda. Duration: 6–26 weeks Frequency varied from five times per week to weekly and classes were 60–90 min. | SF-36, EORTC QLQ-C30, FACT-B, FACT-G, SF-12, NHP | Yoga for cancer survivors results in clinically significant improvements in overall HRQoL, as well as in its mental and emotional domains. |
| Duijts et al[9] | Evaluate the effect of behavioural techniques and physical exercise on psychosocial functioning and HRQoL in patients with breast cancer and survivors | 56 RCTs | >7000 patients with breast cancer, including non-metastatic and metastatic patients during and after treatment Participants' ages were not specified. | Patients during and after treatment | Face to face, online or by telephone, individually or at group level | Behavioural techniques included psychoeducation, problem solving, stress management, CBT, relaxation techniques, social and emotional support. Physical interventions included yoga, self-management exercise protocol, aerobic or resistance exercise training and dance movement. Intervention duration varied from 1 to 56 weeks of 3–56 sessions. | SIP, CARES, ABS, EORTC QLQ-C30, FACT-B, FACT-G, FACT-F, FACT-An, FLIC, SF-12, SF-36, QoL-BC, GHQ, SDS, IFS-CA, VAS | There is no significant effect of behavioural techniques on HRQoL. Physical exercise produced statistically significant but moderate effects on HRQoL. |

Continued

**Table 3** Continued

| Review | Aims of review | Primary studies (n) | Participants | Definition of 'survivor' | Setting | Intervention, duration and frequency | Outcome—QoL measures | Narrative findings |
|---|---|---|---|---|---|---|---|---|
| Ferrer et al[19] | To examine the efficacy of exercise interventions in improving quality of life in cancer survivors, as well as features that may moderate such effects | 78 studies/43 RCTs | 3629 participants: 54% breast cancer, 8% prostate cancer, 2% colorectal cancer, 1% each featured endometrial, head–neck, lymphoma and ovarian cancer survivors, and 32% included mixed diagnosis 2432 patients participated in the RCTs. Mean age was 55 years. | Survivor was defined as post diagnosis. | Supervised or unsupervised | Interventions were designed to affect exercise behaviour by comparing low versus high exercise intensity. 36% used trained intervention leaders; 56% featured supervised exercise sessions. The mean level of targeted aerobic METs was 4.2 (SD=2.2), and the mean level of targeted resistance METs was 2.5 (SD=2.2). Duration: 8–26 weeks. The mean length of intervention session was 51.1 min and the mean number of sessions per intervention was 22.8. | EORTC QLQ-30, SF-36, FACTIT, Quality of Life Index, FACT-G, FACT-An, FACT-B, FACT H&N, FACT-P, FLIC, CARES-SF, Rotterdam QoL, WHOQOL-BREF | There was a positive effect of physical interventions on QoL, sustained for delayed follow-up assessment. Efficacy increased as the length of intervention decreased, and if exercise was supervised. Targeted aerobic intensity significantly predicted QoL improvements as a quadratic trend. Targeted aerobic METs predicted intervention efficacy. Number of sessions, targeted resistance METs, training of facilitators and inclusion of flexibility content were not significantly related to QoL outcomes. |
| Fong et al[10] | To systematically evaluate the effects of physical activity in adult patients after completion of main treatment related to cancer | 34 RCTs | 3769 participants; 65% included breast cancer only, 9% colorectal cancer only, 3% endometrial cancer only and 27% mixed diagnosis. Mean age range: 39–74 years | Patients who have completed their main cancer treatment but might be undergoing hormonal treatment | Face to face | Exercise interventions included aerobic exercise, resistance or strength training. Duration: 3–60 weeks Frequency ranged from daily to once a week. | FACT-G, FACT-B, FACT-C, EORTC, SF-36 | Physical activity was shown to be associated with clinically important positive effects on quality of life. Aerobic plus resistance training was significantly more effective than aerobic training alone on general QoL. |

Continued

**Table 3** Continued

| Review | Aims of review | Primary studies (n) | Participants | Definition of 'survivor' | Setting | Intervention, duration and frequency | Outcome—QoL measures | Narrative findings |
|---|---|---|---|---|---|---|---|---|
| Fors et al[24] | To determine the effectiveness of psychoeducation, CBT and social support interventions used in the rehabilitation of patients with breast cancer | 18 RCTs | 3272 patients with breast cancer, during and post treatment Age range not specified | Patients who have finished surgery and adjuvant treatment | Online, face to face or by telephone or by using print material, individually or in a group | Psychoeducation, CBT and social and emotional support Duration ranged from 2 weeks to 6 months. | FACT-B, FACT-G, EORTC-QLQ-C30, QoL-BC, QLI, EuroQoL-5D, QoQ-C33 Global | Psychoeducation showed inconsistent results during and after primary treatment. CBT after primary treatment (6–12 weeks) led to improved QoL. CBT during primary treatment had inconsistent results. |
| Galvão and Newton[13] | To present an overview of exercise interventions in patients with cancer during and after treatment and evaluate dose-training response considering type, frequency, volume and intensity of training along with physiological outcomes | 26 studies/9 RCTs | 1186 patients with mixed cancer during and post treatment 458 patients participated in the RCTs. Age range: 14–65 years | Patients during and after treatment | Face to face | Exercise interventions included a cardiovascular exercise programme and mixed training (cardio, resistance and flexibility exercises). Intensity level when provided was described as between 60% and 80% maximum heart rate. Programme duration was 4–28 weeks. Frequency ranged from twice a week to five times per week. | Modified Rotterdam QoL Survey | Contemporary resistance training provides anabolic effects that counteract side effects of cancer treatments to improve quality of life. |
| Gerritsen and Vincent[20] | To evaluate the effectiveness of exercise in improving QoL in patients with cancer, during and after treatment | 16 RCTs | 1845 patients with mixed, breast, lymphoma, colorectal, prostate and lung cancer Aged: 18–79 years | Patients during or after treatment | Home-based or outdoors, supervised or unsupervised | Exercise modalities included walking, cycling, strength training, swimming, stability training and elliptical training ranging from twice a week to five times a week. The duration ranged from 3 weeks to 16 months. | EORTC-QLQ, FACT-An, FACT-B, FACT-C, FACT-G, FACT-P, SF-36, MCS/PCS | Exercise has a direct positive impact on the QoL of patients with cancer, during and following medical intervention. |

Continued

**Table 3** Continued

| Review | Aims of review | Primary studies (n) | Participants | Definition of 'survivor' | Setting | Intervention, duration and frequency | Outcome—QoL measures | Narrative findings |
|---|---|---|---|---|---|---|---|---|
| Huang et al[27] | Meta-analysis to evaluate the benefits of mindfulness-based stress reduction on psychological distress among breast cancer survivors | 9 studies/4 RCTs | 964 breast cancer survivors 812 patients participated in the RCTs Mean age range: 49–57.5 | Women diagnosed with breast cancer | Setting not specified | 8-week mindfulness-based stress reduction programme One study used a 6-week formula. | FACT-B | Mindfulness-based stress reduction programmes showed a positive effect in improving psychological function and overall QoL of breast cancer survivors. |
| Khan et al[8] | To assess the effects of organised multidisciplinary rehabilitation during follow-up in women treated for breast cancer | 2 RCTs | 262 patients with breast cancer after treatment All women were older than 49 years except for two. | At least 12 months after completion of definitive cancer treatment | Group-based inpatient programme or inpatient programme together with a home-based programme | Multidisciplinary rehabilitation programme incorporating medical input, psychology and physiotherapy or psychology-based education, exercise, peer support group activity and medical input Duration: 3–10 weeks of 3 sessions per week | Local QoL measure, EORTC QLQ-C30 | There was 'low level' evidence that multidisciplinary rehabilitation can improve QoL over 12 months. It was not possible to suggest optimal frequency, or one type of intervention over another. |
| McAlpine et al[15] | To examine the evidence-based literature surrounding the use of online resources for adult patients with cancer | 14 studies/9 RCTs | 2351 patients with lung, prostate, breast, head and neck and mixed cancer The sample size for the RCTs was 1121 patients and their mean age ranged from 49.5 to 67.2 years. | Survivors are defined as patients who have had a cancer diagnosis in the past, including those currently receiving active treatment, those in remission or cured, and those who are in the terminal stages of disease. | A variety of online platforms were used, including email, online educational resources, online support groups or message boards, cancer information websites and interactive websites. | Three interventions: (1) linking patients to their treating team of clinicians, (2) connecting patients with each other, (3) educational resources Duration: 4 weeks to 12 months | FACT-B, SF-12, EORTC QLQ-C30, EQ-5D, EPIC-26, 15DHRQoL, bespoke QoL measure | The overall benefit of online interventions for patients with cancer is unclear. Although there is significant promise, the few interventions that have been rigorously analysed demonstrate mixed efficacy, often of limited duration. |

Continued

**Table 3** Continued

| Review | Aims of review | Primary studies (n) | Participants | Definition of 'survivor' | Setting | Intervention, duration and frequency | Outcome—QoL measures | Narrative findings |
|---|---|---|---|---|---|---|---|---|
| Mewes et al[18] | To systematically review the evidence on the effectiveness of multidimensional rehabilitation programme for cancer survivors and to critically review the cost-effectiveness studies of cancer rehabilitation | 16 studies originated from 11 trials (11 RCTs, 3 pretest-post-test, 1 quasi-experimental, 1 longitudinal) | 2175 patients with mixed cancer, predominantly breast RCTs included from 21 to 199 participants Age range not specified | Patients with any type of cancer who finished primary treatment with an expected survival duration of at least 1 year Hormone therapy could still be ongoing. | Face to face in an inpatient setting | Multidimensional rehabilitation defined as consisting of two or more rehabilitation interventions directed at the ICF dimensions Interventions typically included exercise, CBT, psychotherapy, education and return to work interventions. Programme duration: 4–15 weeks | EORTC QLQ-C30, RAND-36, FACT-G, FACT-B, SF-12 | Effect sizes for QoL were in the range of −0.12 (95% CI −0.45 to −0.20) to 0.98 (95% CI 0.69 to 1.29). Multidimensional and monodimensional interventions were equally effective. |
| Mishra et al[12] | To evaluate the effectiveness of exercise on overall HRQoL and HRQoL domains among adult post-treatment cancer survivors | 40 trials/38 RCTs | 3694 patients with mixed cancer during and post-treatment were randomised. Over 50% included patients with breast cancer only. Mean age range: 39–68 years | Participants who have completed treatment | Settings included a gym, community centre, yoga studio, or university or hospital facility. Home-based interventions were included. | Exercise was defined as physical activity causing an increase in energy expenditure in a systematic manner in terms of frequency, intensity and duration. Interventions included prescribed, active exercise formats of aerobic, resistance, stretching or aerobic/resistance combinations. Some interventions included modules in psychological or behavioural education. Duration ranged from 3 weeks to 1 year. Frequency varied from daily to once per week. Sessions lasted from 20 to more than 90 min. | EORTC QLQ-C30, FACT-G, FACT-B, FACT-F, FACT-An, FACT-Lym, FACIT-F, CARES-SF, QoL Index, SF-36, Neck Dissection Impairment Index for QoL for head and neck cancer survivors | Exercise has a positive impact on QoL with improvements in global QoL. |

Continued

**Table 3** Continued

| Review | Aims of review | Primary studies (n) | Participants | Definition of 'survivor' | Setting | Intervention, duration and frequency | Outcome—QoL measures | Narrative findings |
|---|---|---|---|---|---|---|---|---|
| Osborn et al[17] | To investigate the effects of CBT and patient education (PE) on commonly reported problems (ie, depression, anxiety, pain, physical functioning and quality of life) in adult cancer survivors | 15 RCTs | 1492 patients with mixed cancer Age range: 18–84 years | Defined as beyond the time of diagnosis | In a group or individually, face to face | Interventions included group or individual CBT, PE. CBT intervention duration ranged from 3 to 55 weeks. Frequency varied from 1 hour per week to 2 hours per week. PE duration ranged from one 20 min session to six weekly 1 hour sessions. | FACT | QoL was improved at short-term and long -term follow-up after CBT. PE was not related to improved outcomes. Individual interventions were more effective than group. |
| Smits et al[21] | To evaluate the effectiveness of lifestyle intervention in improving QoL of endometrial and ovarian cancer survivors | 8 studies/3 RCTs | 413 survivors of endometrial and ovarian cancer were included in the analysis. 153 survivors were included in the RCTs. Age range was not specified. | Adults diagnosed with endometrial cancer having completed primary treatment (surgery, chemotherapy or radiotherapy) | Home-based, individually or group-based | Physical activity, behavioural change, nutritional, counselling interventions The duration varied from 12 weeks to 12 months. | FACT-G, FACT-F, FACT-O, SF-36 and QLACS | The review did not show improvements in global QoL. The authors concluded that lifestyle interventions have the potential to improve QoL in this population. |
| Spark et al[25] | To determine the proportion of physical activity and/or dietary intervention trials in breast cancer survivors that assessed postintervention maintenance of outcomes, the proportion of trials that achieved successful postintervention maintenance of outcomes, and the sample, intervention and methodological characteristics common among trials that achieved successful postintervention maintenance of outcomes | 16 studies originated from 10 RCTs | 1536 breast cancer survivors during or after treatment Age range not specified | Not specified | Interventions included face-to-face contact, printed information and telephone counselling or home-based delivery. | Interventions were described as physical activity and/or dietary behaviour change aiming to increase aerobic fitness, strength and physical activity. Most interventions lasted 1–4 months, with some lasting longer than 6 months. | Measures not specified | More research is needed to identify the best ways of supporting survivors to make and maintain these lifestyle changes. QoL-specific outcomes from three studies were not reported. |

Continued

**Table 3** Continued

| Review | Aims of review | Primary studies (n) | Participants | Definition of 'survivor' | Setting | Intervention, duration and frequency | Outcome—QoL measures | Narrative findings |
|---|---|---|---|---|---|---|---|---|
| Spence et al[16] | To summarise the literature on the health effects of exercise during cancer rehabilitation and to evaluate the methodological rigour of studies in this area | 13 studies originated from 10 trials, 4 of which were RCTs | 327 patients with mixed cancer, mostly patients with breast cancer The sample size for the RCTs was 245 patients and their mean age ranged from 18 to 65 years. | Patients who had recently completed treatment and had reported no plans for additional treatment 'Recently completed' was defined as having completed treatment no more than 12 months prior to enrolment. | Interventions were either supervised exercise programmes or home-based, unsupervised exercise programmes. One study employed exercise physiologists to prescribe individually tailored exercise programmes. | Most interventions were aerobic or resistance-training exercise programmes. Most studies prescribed cycling or walking ergometers for the aerobic component. Studies incorporating resistance training prescribed either exercises using machines or resistance bands. Duration varied from 2 weeks to 14 weeks with a frequency of daily exercise to two or three sessions per week. | Cancer Rehabilitation Evaluation System | The findings from this review suggest that exercise can provide a variety of benefits for cancer survivors during the rehabilitation period, including an improved QoL. |
| Zachariae and O'Toole[22] | To evaluate the effectiveness of expressive writing for improving psychological and physical health in patients with cancer and survivors | 16 RCTs | 1797 patients with cancer or survivors Breast cancer, ovarian, renal, prostate, colorectal and mixed cancers Age range not specified | Not specified | Lab or home-based | Expressive writing interventions requiring participants to disclose their emotions in sessions The duration of the intervention ranged from 3 to 4 sessions, which were daily, weekly or biweekly. | FACT-B, FACT-G, FACT-BMT, QLQ-C30 | The review did not support the general effectiveness of expressive writing in patients with cancer and survivors. |

Continued

**Table 3** Continued

| Review | Aims of review | Primary studies (n) | Participants | Definition of 'survivor' | Setting | Intervention, duration and frequency | Outcome—QoL measures | Narrative findings |
|---|---|---|---|---|---|---|---|---|
| Zeng et al[26] | To examine the effectiveness of exercise intervention on the quality of life of breast cancer survivors | 25 studies included in the qualitative synthesis, 19 studies included in meta-analysis 22 RCTs | 1073 patients with breast cancer aged 18 years or over | Individuals who had completed active cancer treatment | Face to face, by telephone | Interventions included any type of exercise—aerobic, resistance or combination of aerobic and resistance, yoga, tai chi, aerobic and strength training, aerobic and resistance training and stretching. The duration of the intervention ranged from 4 to 52 weeks. Time per session varied from 15 to 90 min, 1–5 times per week. | Generic QoL measures: SF-36, FACT-G, EORTC-QLQ-C30 Cancer site-specific QoL measures: FACT-B, EORTC QLQ BR23 | The review found consistent positive effects of exercise interventions in overall QoL and certain QoL domains. There was a small to moderate effect of interventions on site-specific QoL. Single type of exercise intervention, general aerobic, yoga or tai chi had significant differences in QoL score changes. |

15DHRQoL, 15 Dimensional Health Related Quality of Life; ABS, Affects Balance Scale; CARES, Cancer Rehabilitation Evaluation System; CARES-SF, Cancer Rehabilitation Evaluation System Short Form; CBT, cognitive behavioural therapy; EORTC QLQ BR23, European Organisation for Research and Treatment of Cancer Quality of Life Questionnaire - Breast Cancer Module; EORTC QLQ-C30, European Organisation for Research and Treatment of Cancer Quality of Life Questionnaire Core 30; EORTC QLQ-C33, European Organisation for Research and Treatment of Cancer Quality of Life Questionnaire C33; EPIC, Expanded Prostate Cancer Index Composite; EPIC-26, Expanded Prostate Cancer Index Composite Short Form; EQ-5D, EuroQol-5D 'feeling thermometer'; FACIT-F, Functional Assessment of Chronic Illness—Fatigue; FACIT-Sp, Functional Assessment of Chronic Illness—Spiritual Well-Being; FACT H&N, Functional Assessment of Cancer Therapy—Head & Neck; FACT-An, Functional Assessment of Cancer Therapy—Anaemia Scale; FACT-B, Functional Assessment of Cancer Therapy—Breast Cancer; FACT-BMT, Functional Assessment of Cancer Therapy—Bone Marrow Transplant; FACT-C, Functional Assessment of Cancer Therapy—Colorectal; FACT-F, Functional Assessment of Cancer Therapy—Fatigue; FACT-G, Functional Assessment of Cancer Therapy—General; FACTIT, Functional Assessment of Chronic Illness Therapy; FACT-Lym, Functional Assessment of Cancer Therapy—Lymphoma; FACT-O, Functional Assessment of Cancer Therapy—Ovarian; FACT-P, Functional Assessment of Cancer Therapy—Prostate; FLIC, Functional Living Index for Cancer; GHQ, General Health Questionnaire; HRQoL, Health-related quality of life; ICF, International Classification of Functioning, Disability and Health; IFS-CA, Inventory of Functional Status—Cancer; MCS/PCS, Mental Component Score/Physical Component Score; MET, Metabolic Equivalents of Task; NHP, Nottingham Health Profile; PCa-QoL, Prostate Cancer Quality of Life Instrument; QLACS, Quality of Life in Adult Cancer Survivors; QLI, Quality of Life Index; QLQ-PR25, European Organisation for Research and Treatment of Cancer (EORTC)-Qualify of Life Questionnaire-Prostate Cancer Module; QoL, quality of life; QoL–BC, Quality of Life Questionnaire—Breast Cancer; QoQ-C33, European Organisation for Research and Treatment of Cancer (EORTC)-Qualify of Life Questionnaire Core 33; RAND-36, 36-Item Short Form Health Survey; RCT, randomised controlled trial; SDS, Symptom Distress Scale; SF-12, Medical Outcomes Study Short-Form Health Survey 12; SF-36, Medical Outcomes Study Short-Form Health Survey 36; SIP, Sickness Impact Profile; UCLA-PCI, University of California, Los Angeles, Prostate Cancer Index; VAS, Visual Analogue Scale; WHOQOL–BREF, WHO Health Organisation Quality of Life Assessment.

yoga[11 14 23]; four reviews were of psychosocial or behavioural interventions[9 17 24 27]; and one review focused on online interventions including connecting patients and online education (see tables 3 and 4).[15] One review compared multidimensional versus monodimensional interventions,[18] and one tested multidisciplinary rehabilitation models.[8] Finally, one review focused on the effects of expressive writing.[22] The duration and frequency of the interventions varied greatly from a single 20 min session[17] to 60 weekly sessions.[10]

The most common components of physical interventions were aerobic exercise[9 10 12 13 16 19 26] and resistance/strength training.[9 10 12 13 16 26] Psychological education[8 9 17 18 24] and cognitive behavioural therapy (CBT)[9 17 18 24] were the most commonly used psychological and educational interventions. Peer support was often used as a psychological and a behavioural intervention.[8 9 15] Components of the interventions were thematically organised into two groups (see table 4 for a more detailed itemisation): biological or physical actions (19 types of activity or diet change), and psychological, behavioural or educational (24 types of intervention about mind and body, including CBT, mindfulness-based stress reduction, psychosexual therapy, supporting existing coping methods, emotional support, relaxation, psychotherapy and psychosocial therapy, and interventions focusing on social support, guided imagery, self-management, use of peer support, bibliotherapy, telephone and web-based interventions, and return to work interventions).

### Overall effectiveness of interventions: meta-analysis findings

Meta-analyses were reported in 11 reviews and the effect sizes (as reported in the original reviews) are tabulated (table 5). Of six publications providing meta-analyses of physical activity (not including yoga), all found convincing positive associations for studies testing response between 1 and 26 weeks post-treatment. Long-term effects were not tested by all, although Fong et al and Zeng et al did show persistent effects at 6 months and a year, respectively.[10 26] One review[19] showed uncertain outcomes at 3–6 months, although shorter and longer term outcomes were favourable. This review showed equivocal effects when the intervention group was compared with the control group, once adjusted for QoL and covariates at baseline. The two meta-analyses of yoga interventions showed positive effects,[11 23] as did a review of CBT.[17] There was no evidence of benefit in QoL following patient education[17] and behavioural interventions.[9]

Two reviews reported effect sizes from individual studies but did not undertake meta-analyses.[18 24] Mewes et al's[18] review of multidimensional rehabilitation included 10 studies, 9 of which had global QoL outcomes; of these, 7 showed benefit, with effect sizes ranging from 0.04 to 0.99 (no CIs reported). Fors et al's[24] review included six RCTs only, four of which included a QoL measure; two of these showed positive effect sizes (ranging from 0.56, 95% CI 0.09 to 1.03; 0.63, 95% CI: 0.11 to 1.18); one showed improved and one a worsening of QoL as a non-standardised mean score. Five reviews[8 13 15 16 25] did not report meta-analyses or effect sizes; mostly these provided mean change scores or narrative statements. On the whole these gave a mixed picture, often resorting to subgroup analysis by cancer type or different dimensions of QoL.

### Physical activity: summary findings

Cramer et al's[23] high-quality review of 6–12 weeks of yoga in patients with breast cancer showed a large increase in general QoL, a finding that was consistent with reviews by Buffart et al[11] and Culos-Reed et al,[14] which scored lower on the AMSTAR. Mishra et al's[12] high-quality review of people with multiple cancers, 50% of whom had breast cancer, found that physical activity had a positive effect on global QoL at 3 and 6 months of follow-up, as did Smits et al's high-quality review of endometrial cancer and Gerritsen and Vincent's moderate-quality review of mixed cancers.[20 21] Fong et al's[10] high-quality review of breast cancer, colorectal, endometrial and mixed cancers similarly found physical interventions improved general QoL on average at 13 weeks of follow-up (range 3–60 weeks). Bourke et al's[28] review of prostate cancer found personalised lifestyle interventions helpful, and McAlpine et al's[15] review of mixed cancers including prostate found benefit of activity following medication treatment.

There was inconsistency across the reviews with regard to the types of exercise interventions that were most effective. Fong et al[10] found aerobic plus resistance training to be significantly more effective than aerobic training alone on many aspects of QoL. However, Zeng et al's[26] moderate-quality review suggested that single types of exercise interventions (general aerobic, yoga or tai chi) were more effective at increasing QoL at 4–52 weeks after intervention; half of the studies assessed interventions between 8 and 12 weeks. Duijts et al's[9] study of patients with breast cancer found only small effects of physical activity on QoL (at 8–26 weeks after intervention), and Spence et al's[16] study of mixed but mostly patients with breast cancer reported evidence that physical activity improved overall QoL, but only four of ten trials maintained the intervention and only a fifth of trials seemed to assess outcome at 3 months and beyond. Zeng et al's[26] review of patients with breast cancer found small but positive benefits of physical activity on overall QoL. Galvão and Newton's[13] review of mixed cancers gave preliminary evidence of positive benefits on a Modified Rotterdam QoL measure, but no overall effects were reported. However, Spark et al's[25] review of patients with breast cancer showed that the impact of physical activity on QoL was not convincing. Although Spark et al did not report effect sizes, two of the studies in that review included QoL measures, both of which reported effect sizes in the original papers: one showed positive benefits on Functional Assessment of Cancer Therapy—General (FACT-G) and Functional Assessment of Cancer Therapy—Breast Cancer at 8 months (effect sizes 9.8–13.4), but not at 24 months of follow-up; the other showed no significant effects on FACT-G overall, but when the cancer-specific FACT-G

**Table 4**  Components of the interventions by study

| | Cramer et al[23] | Fong et al[10] | Buffart et al[11] | Khan et al[8] | Mishra et al[12] | Culos-Reed et al[14] | Bourke et al[28] | Duijts et al[9] | Ferrer et al[19] | Fors et al[24] | Galvão and Newton[13] | Gerritsen and Vincent[20] | Huang et al[27] | McAlpine et al[15] | Mewes et al[18] | Osborn et al[17] | Smits et al[21] | Spark et al[25] | Spence et al[16] | Zachariae and O'Toole[22] | Zeng et al[26] |
|---|---|---|---|---|---|---|---|---|---|---|---|---|---|---|---|---|---|---|---|---|---|
| **Physical** | | | | | | | | | | | | | | | | | | | | | |
| Aerobic | | ● | | | | | | | | | | | | | | | | | | | ● |
| Aerobic and resistance | | | | | ● | | ● | ● | ● | | ● | | | | | | ● | | ● | | ● |
| Resistance | | | | | ● | | ● | | | | ● | | | | | | | | | | ● |
| Aquatic exercise | | | | | ● | | | | | | | | | | | | | | | | |
| Cardiovascular programme | | | | | | | | | | | ● | | | | | | | | | | ● |
| Cycling | | | | | ● | | | | | | ● | ● | | | | | | | ● | | |
| Dance movement | | | | | | | | ● | | | | | | | | | | | | | |
| Enhanced standard care | | | | | | | ● | | | | | | | | | | | | | | |
| Exercise not specified | | | | ● | | | | | | | | | | | ● | | ● | ● | | | |
| Expressive writing | | | | | | | | | | | | | | | | | | | | ● | |
| METs targeted | | | | | | | | | ● | | | | | | | | | | | | |
| Dietary intervention | | | | ● | | | ● | | | | | | | | | | ● | ● | | | |
| Pilates | | | | | ● | | | | | | | | | | | | | | | | |
| Resistance/strength training | | ● | | | ● | | | ● | | | ● | ● | | | | | ● | | ● | | ● |
| Running | | | | | ● | | | | | | | | | | | | | | | | |
| Self-management exercise | | | | | | | | ● | | | | | | | | | | | | | |
| Stretching/flexibility exercises | | | | | | | | | | | ● | | | | | | | | ● | | ● |
| Swimming | | | | | | | | | | | | ● | | | | | | | | | |
| Tai chi | | | | | ● | | | | | | | | | | | | | | | | ● |
| Treadmill | | | | | | | | | | | | | | | | | | | ● | | |
| Walking | | | | | ● | | | ● | | | ● | ● | | | | | | | ● | | ● |
| Weight training | | | | | | | | ● | | | | | | | | | | | | | |
| Yoga/meditation | ● | | ● | | ● | ● | | ● | | | | | | | | | | | | | ● |
| Qigong | | | | | ● | | | | | | | | | | | | | | | | |
| **Psychological, educational and behavioural** | | | | | | | | | | | | | | | | | | | | | |
| Body mind | | | | | | | | ● | | | | | | | | | | | | | |
| Cognitive behavioural stress therapy | | | | | | | ● | ● | | | | | | | | ● | | | | | |
| Cognitive behavioural therapy | | | | | | | | ● | | ● | | | | | ● | ● | | | | | |

**Table 4** Continued

| | Cramer et al[23] | Fong et al[10] | Buffart et al[11] | Khan et al[8] | Mishra et al[12] | Culos-Reed et al[14] | Bourke et al[28] | Duijts et al[9] | Ferrer et al[19] | Fors et al[24] | Galvão and Newton[13] | Gerritsen and Vincent[20] | Huang et al[27] | McAlpine et al[15] | Mewes et al[18] | Osborn et al[17] | Smits et al[21] | Spark et al[25] | Spence et al[16] | Zachariae and O'Toole[22] | Zeng et al[26] |
|---|---|---|---|---|---|---|---|---|---|---|---|---|---|---|---|---|---|---|---|---|---|
| Cognitive G therapy | | | | | | | | • | | | | | | | | | | | | | |
| Combined psychosexual | | | | | | | | • | | | | | | | | | | | | | |
| Comprehensive coping strategy | | | | | | | | • | | | | | | | | | | | | | |
| Coping skills | | | | | | | | • | | | | | | | | | | | | | |
| Emotional support | | | | | | | | • | | • | | | | | | | | | | | |
| Group therapy | | | | | | | • | • | | | | | | | | • | • | | | | |
| Guided imagery | | | | | | | | • | | | | | | | | | | | | | |
| Image consultant | | | | • | | | | | | | | | | | | | | | | | |
| Mindfulness-based stress reduction programme | | | | | | | | | | | | | • | | | | | | | | |
| Motivational interviewing | | | | | | | | | | | | | | | | | | | | | |
| Problem-solving training | | | | | | | | • | | | | | | | | | | | | | |
| Progressive relaxation training | | | | | | | | | | | | | | | | | | | | | |
| Psychotherapy | | | | | | | | | | | | | | | • | | | | | | |
| Psychosocial therapy | | | | | | | | • | | | | | | | | | | | | | |
| Return to work interventions | | | | | | | | | | | | | | | • | | | | | | |
| Social support | | | | | | | | • | | • | | | | | | | | | | | |
| Stress management | | | | | | | | • | | | | | | | | | | | | | |
| Health education | | | | | | | | • | | | | | | | • | • | | | | | |
| Psychological education | | | | • | | | • | • | | • | | | | | • | • | | | | | |
| Peer support | | | | • | | | | • | | | | | | • | | | | | | | |
| Mode of delivery | | | | | | | | | | | | | | | | | | | | | |
| Compact discs/manuals/videos | | | • | | | | • | | | • | | | | | | | | | | | |
| Face to face | | | | • | • | | • | • | | | | | | | | | • | • | | | |
| Home-based | | | • | • | • | | | • | | | | | | | | | • | • | • | | |
| Inpatient setting | | | | • | | | | | | | | | | | • | | | | | | |
| Multidisciplinary rehabilitation programme | | | | • | | | • | | | | | | | | • | | | | | | |
| Printed information | | | • | | | | | | | | | | | | | | | • | | | |

Continued

**Table 4** Continued

| | Cramer et al[23] | Fong et al[10] | Buffart et al[11] | Khan et al[8] | Mishra et al[12] | Culos-Reed et al[14] | Bourke et al[28] | Duijts et al[9] | Ferrer et al[19] | Fors et al[24] | Galvão and Newton[13] | Gerritsen and Vincent[20] | Huang et al[27] | McAlpine et al[15] | Mewes et al[18] | Osborn et al[17] | Smits et al[21] | Spark et al[25] | Spence et al[16] | Zachariae and O'Toole[22] | Zeng et al[26] |
|---|---|---|---|---|---|---|---|---|---|---|---|---|---|---|---|---|---|---|---|---|---|
| Support from nurse or voluntary organisations | | | | | | | ● | ● | | ● | | | | | | | | | | | |
| Telephone | ● | | ● | | | | | | | | | | | | | | | ● | | | ● |
| Web-based | | | | | | | ● | ● | | ● | | | | ● | | | | | | | |

MET, metabolic equivalents of task.

was assessed at 6-month follow-up, there was benefit (4.9, 0.2–9.6). Ferrer et al's[19] study of breast, prostate, endometrial, head and neck, ovarian cancers and lymphoma found small but positive effects of exercise at long-term follow-up on multiple measures of QoL. The efficacy of the interventions appeared greater with shorter duration treatments, and if exercise was supervised. Aerobic intensity predicted improvements in QoL.

### Psychological and behavioural interventions: summary findings

Only one of the reviews of psychological and behavioural interventions was classified as high quality: Huang et al's[27] meta-analysis of patients with breast cancer showed that mindfulness-based stress reduction programmes had a significant effect in improving overall QoL. Duijts et al's[9] review, on the other hand, concluded that behavioural techniques such as problem solving, stress management and CBT did not significantly improve health-related QoL. Nevertheless, Fors et al's[24] review of patients with breast cancer showed CBT improved QoL. No meta-analysis or overall effect sizes were reported due to heterogeneity. Further support for CBT came from Osborn et al's[17] review of group and individually delivered CBT for mixed cancers; individual interventions were more effective than group-based treatment. CBT showed both short-term[24] and long-term improvements in QoL.[17] Five primary papers in one review assessed the effect of social and emotional support as an intervention, four of them finding no effect, and one reporting a significant improvement in QoL on one measure.[24] There was no evidence that psychosocial education increased QoL.[17 24]

### Multidimensional and multidisciplinary rehabilitation

Khan et al's[8] high-quality review of patients with breast cancer included just two studies, only one of which provided low-level evidence that multidisciplinary rehabilitation improved participation and social activities. The other showed no significant effects. Mewes et al's[18] moderate-quality review of breast and other cancers treated by inpatient multidisciplinary rehabilitation demonstrated no differences between multidimensional and single-dimension interventions, with benefits of both on physical outcomes. Bourke et al's[28] review of prostate cancer survivors examined the effectiveness of multidisciplinary approaches based on findings from three primary studies. They concluded that such interventions showed small benefits for QoL, typically when they involved a smaller number of health professionals, thus allowing more focused tailoring of the interventions.

### Intervention modality

The effectiveness of online educational interventions was unclear. McAlpine et al's[15] review of lung, prostate, head and neck and a smaller number of mixed cancers showed equivocal findings. There were benefits to online education and message boards, but mixed effects for interactive websites, and worse outcomes from one study on email interventions. One interesting review was of expressive

**Table 5** Reported effect size from meta-analyses in reviews

| Authors | Intervention | Type of effect size reported | Reported effect size | Overall finding |
|---|---|---|---|---|
| Buffart et al[11]* | Yoga | SMD (7 studies)<br>General QoL | 0.37, 0.11 to 0.62 | + |
| Cramer et al[23]* | Yoga | SMD (4 studies)<br>Global QoL | 0.62, 0.04 to 1.21 | + |
| Ferrer et al[19]*‡ | Exercise | SMD (78 studies) | | |
| | | All intervention groups (immediate FU) | 0.34, 0.24 to 0.43 | + |
| | | Intervention versus control, adjusted for baseline differences | 0.24, 0.12 to 0.35 | + |
| | | Delayed FU | | |
| | | All intervention groups | 0.42, 0.23 to 0.61 | + |
| | | Intervention versus control adjusted for baseline | 0.20, −0.058 to 0.46 | + |
| Fong et al[10] | Exercise | 2 studies | 3.4, 0.4 to 6.4 | + |
| | | 9 studies | 22.1, 16.8 to 27.4 | + |
| Gerritsen and Vincent[20] | Exercise | SMD: intervention versus control | 5.55, 3.19 to 7.9 | + |
| Mishra et al[12]* | Exercise | SMD: baseline to after intervention (11 studies) | 0.48, 0.16 to 0.81 | + |
| | | Follow-up of 3–6 months (181 participants) | 0.14, −0.38 to 0.66 | − |
| | | 6-month follow-up (115 participants) (2 studies) | 0.46, 0.09 to 0.84 | + |
| Zeng et al[26] | Exercise | Standardised mean difference (overall) (6 studies) | 0.70, 0.21 to 1.19 | + |
| | | Cancer-specific (10 studies) | 0.38, 0.03 to 0.74 | + |
| Duijts et al[9] | Exercise | SMD (or Hedges' g for small sample size, with adjustment) (27 studies) | 0.298, 0.117 to 0.479, P=0.001 | + |
| | Behavioural intervention | | 0.045, −0.044 to 0.135, P=0.322 | Uncertain |
| Osborn et al[17] | CBT | SMD overall (11 studies) | 0.91, 0.38 to 1.44, P<0.01 | + |
| | | Short term (<8 weeks) | 1.45, 0.43 to 2.47 | + |
| | | Long term (>8 weeks) | 0.26, 0.06 to 0.46 | + |
| | | Individual CBT (7 studies) | 0.95, −0.367 to 1.536 | + |
| | | Individual versus group CBT (1 study) | 0.37, −0.02 to 0.75 | Uncertain |

Continued

**Table 5** Continued

| Authors | Intervention | Type of effect size reported | Reported effect size | Overall finding |
|---|---|---|---|---|
|  | Patient education | (1 study) | −0.04, −0.38 to 0.29 | − |
| Smits et al[21] | Lifestyle interventions | SMD |  | + |
|  |  | 3 months | 1.16, −5.91 to 8.23 |  |
|  |  | 6 months | 2.48, −4.63 to 9.58 |  |
| Zachariae and O'Toole[22] | Expressive writing | Hedges's g | 0.09, −0.5 to 0.24 | + |

*Reviews rated as high quality.
†Random effects assumption.
‡Findings sustained for random or fixed effects, random effects reported.
CBT, cognitive behavioural therapy; FU, follow up; QoL, quality of life; SMD, Standardised Mean Difference.

writing interventions, but this found no benefit on QoL, although small effects would be undetected.[22] Individuals with low levels of emotional support appeared to benefit more than others.

### Adverse effects

Five reviews[11 12 15 23 26] included reports of adverse events. Of four studies in Buffart et al's[11] review, one reported back spasm in a yoga class in a patient with a history of back problems. In Cramer et al's[23] review of three studies reporting adverse events, there was one adverse event (back spasm) in 138 patients. McAlpine et al's[15] review included two studies that reported adverse effects of online support groups. One of these reported transient helplessness, anxiety, confusion and depression at 6 months, while the other showed poorer QoL despite high levels of reported satisfaction. Zeng et al's[26] review of 25 trials found one study with reports of exercise-related lymphoedema. In Mishra et al's[12] review, six studies reported adverse effects including lymphoedema, gynaecological complications and influenza in the exercise group. One study reported back, knee and hip problems. Three participants in one study reported thrombosis and infection following exercise interventions. Another study found hip pain, sciatica, arm discomfort (n=4), knee discomfort (n=10), ankle discomfort (n=3), and foot discomfort (n=8) with asymptomatic ischaemia and conduction problems on ECG. A further study reported lung metastases, pulmonary embolism and palpitations. Another study reported soft tissue injury following exercise, and cholecystitis following stroke. Cancer recurrence, although not a direct effect of interventions, was common and another reason to stop participation in the research.

## DISCUSSION

### Main findings

Twenty-one reviews were included and showed a lack of definitive and consistent evidence across 465 primary studies, of which 362 were RCTs. In part this is explained by substantial variation in study designs and outcome measures used to indicate QoL. All systematic reviews of physical activity demonstrated improved overall QoL, but few studies assessed long-term outcomes beyond 3 months, and even fewer assessed outcomes beyond a year after the intervention. More focused research and a consistent approach are required to explore the effect on the subdomains of QoL.[12] A higher quality review suggests that aerobic plus resistance training provides maximum improvements in QoL.[10] There was more evidence of physical rather than psychological or other types of interventions.

One of the included reviews for psychological or behavioural interventions was of high quality.[27] CBT is effective for improving QoL in the short and long term,[17 24] especially when provided as an individual intervention.[17] There is not much evidence to support

comparative effectiveness of intervention modalities such as group versus individual, monodimensional versus multidimensional or multidisciplinary; further work is needed to examine these different approaches. Given the accessibility of social media and its popularity, the findings that email contact was related to poorer QoL need further investigation; although interactive websites were beneficial, the overall findings about digital interventions were equivocal.

## Limitations

The current review has some limitations in the methodology. Studies not in English and grey literature were not included due to time constraints as the review was undertaken as part of a programme development grant to inform the design of a future research programme application.

We encountered some methodological limitations in included reviews. Some used multiple outcomes and often had a very broad understanding of QoL and used diverse measures of QoL. There was no consistent reporting standard.

We did not consider outcomes such as well-being or the multiple subdomains of QoL to avoid the risk of generating findings due to multiple testing in smaller subsamples in underpowered analyses. Some reviews included few primary papers. We examined the sample sizes of RCTs included in reviews and whether there seemed to be any relationship with AMSTAR ratings. We found no obvious relationship, given AMSTAR scores refer to review quality rather than the quality of or sample size of individual RCTs. A review of primary RCTS might help to better understand and report robust findings from RCTs with large and adequate sample sizes, findings which may otherwise be less visible in a review of reviews.

We found little overlap between reviews (tabulation available on request), reflecting their specific inclusion and exclusion criteria and interest in very specific interventions and cancer types. We did not evaluate the methodological quality or bias of the original studies within each systematic review. Ten reviews planned to assess publication bias; three of these could not perform any specific tests of bias due to small samples.[8 23 27] Consequently seven studies tested for publication bias.[9 10 12 17 19 20 22] Three of these reported that publication bias was not significant.[10 20 22] Four reviews[9 12 17 19] reported significant publication bias suggesting caution in assuming there is definitive evidence for exercise and CBT.

The physical and psychosocial concerns of patients at different time periods of the cancer experience will vary greatly, and interventions effective at one stage may not be suitable for another. Most reviews defined 'survivors' as those who had completed active treatment before the onset of the study.[10 13 14 16 18 19 23 24 26] Some specified a time frame, from immediately after surgery to 15 years after active treatment.[12] One review defined survival as being from diagnosis onwards.[17] Another included terminal stages of cancer.[15] The majority of the reviews incorporated studies combining patients during and post-treatment.[9 11–15 23–25] These differing definitions of living with and beyond cancer make comparison difficult, and a standardised approach to trials and reporting of studies is needed.

Interventions were offered to patients based on their diagnosis of cancer, rather than low QoL, which may have led to underestimation of potential beneficial effects. Future research should consider the effectiveness of interventions targeting people living beyond all types of cancer and with poor overall QoL.

## CONCLUSIONS

Systematic reviews of patients with cancer and their QoL showed that effective interventions included physical activity, CBT and mindfulness-based stress reduction training. Personalised lifestyle interventions showed promise, as did social and emotional support. Educational and information provision appears ineffective, and there were few studies of electronic interventions. Currently, there is no standard study design, outcome selection or reporting convention adopted across these reviews. No single intervention can be recommended to those patients with a poor QoL following cancer treatment as interventions were not targeting poorer QoL, but cancer survivors in general.

**Author affiliations**
[1]Academic Psychological Medicine, Wolfson Institute of Preventive Medicine, Barts and The London School of Medicine and Dentistry, Queen Mary University of London, London, UK
[2]Centre for Psychiatry, Barts and The London School of Medicine and Dentistry, Queen Mary University of London, London, UK
[3]Blizard Institute, National Bowel Research Centre, Queen Mary University of London, London, UK
[4]Department of Colorectal Surgery, The Royal London Hospital, Barts Health NHS Trust, London, UK
[5]University College Hospitals, NHS Foundation Trust and UCLH Biomedical Research Centre, London, UK
[6]Marie Curie Palliative Research Department, Division of Psychiatry, University College Medical School, London, UK
[7]Cancer Research Group, Sheffield Hallam University, Sheffield, UK
[8]Health & Wellbeing, Sheffield Hallam University, Sheffield
[9]Centre for Tumour Biology, Barts Cancer Institute – Queen Mary University of London, London, UK
[10]Department of Psychological Medicine, King's College London, Denmark Hill, King's College, London, UK
[11]Centre for Primary Care and Public Health, Blizard Institute, Barts and The London School of Medicine and Dentistry, London, UK
[12]Wolfson Institute of Preventive Medicine, Barts and The London School of Medicine and Dentistry, Queen Mary University of London, London, UK

**Acknowledgements** We thank Miriam Harris, Adrienne Morgan, and Louisa Smalley for helpful analysis and comments in the design, planning and delivery of the research including this review, and in the construction of SURECAN dissemination plans and the design of a future trial.

**Collaborators** SURECAN Investigators and Research Group.

**Contributors** KB as PI for the review designed the review and prepared the review section for the original grant application, which overall was led by PDW. Input on design was provided by all authors (KB, MD, JD, RR, LJ, LB, AM, TC, MAT, SCT, AK, PDW) and PPI experts (Miriam Harris, Adrienne Morgan, and Louisa Smalley) in steering groups during preparation of the funding application and throughout the project; more specific additional input to design was provided by PDW and SCT. MD

and JD were research fellows employed on the grant, and collected the papers, ran the searches and performed the first extraction under supervision by KB. MD and JD undertook the preliminary charting and extraction. EH (a PPI expert) and EM conducted the AMSTAR ratings and the final data extraction and edited the draft, under the supervision of KB. KB reviewed all data and checked and completed extraction of the data and identified relevant effect estimates, and led on writing the paper, edited consecutive drafts of the MS, and produced the final draft. All authors (KB, EM, EH, MD, JD, RR, LJ, LB, AM, TC, MAT, SCT, AK, PDW) contributed to the reviewing consecutive drafts of the paper for content, the presentation and discussion about the findings, and interpretation at each stage of the review process, as well as the structure of the paper. All authors (KB, EM, EH, MD, JD, RR, LJ, LB, AM, TC, MAT, SCT, AK, PDW) commented on and approved the final version. KB is the corresponding author.

**Funding** This review was funded by NIHR Programme Development Grant: RP-DG-1212-10014.

**Competing interests** None declared.

**Provenance and peer review** Not commissioned; externally peer reviewed.

**Data sharing statement** No additional data are available.

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
