## [Reviewer comments · BMJ Open]

ARTICLE DETAILS

TITLE (PROVISIONAL)	A Review of Systematic Reviews of non-pharmacological interventions to improve quality of life in cancer survivors.
AUTHORS	Duncan, Morvwen; Moschopoulou, Elisavet; Herrington, Eldrid; Deane, Jennifer; Roylance, Rebecca; Jones, Louise; Bourke, Liam; Morgan, Adrienne; Chalder, Trudie; Thaha, Mohamed; Taylor, Stephanie; Korszun, Ania; White, Peter; Bhui, Kamaldeep

VERSION 1 – REVIEW

REVIEWER	W.H van Harten The Netherlands Cancer Institute
REVIEW RETURNED	22-Jan-2017

GENERAL COMMENTS	The paper presents a rapid review of systematic reviews of non pharmacological interventions to improve the well-being and quality of life in cancer survivors. Overall the review approach is systematic and well performed; the method of analysis however leaves some doubts that need addressing. What is meant with "rapid" in the review is not explained. The authors could better format the way they present the results as there are different ways to structure quality of life and wellbeing, either through WHO based formats, EORTC or ASCO scales etc. Further the AMSTAR score is applied in a binary way. Being the corresponding author for one of the studies (Mewes et.al) I doubt how the score was applied and whether a difference of 4 points merits the score of "Low Quality" as these are based on a low score on conflict of interest not being stated and a combined analysis which was obviously not possible due to different interventions and survivor symptoms involved. This raises doubts on the scores of the other papers and the methodological explanation is not very conclusive (page 10). further more it underlines that the results should be presented in a more detailed way, as so many symptoms, phases, and interventions, (mono disciplinary, multidisciplinary, combined) can be applied. The passages on overall effectiveness suffer from the same issues and as the high/low quality label is presented consequently, this seems to "denounce" certain studies and in addition looking for aggregate scores is not very satisfying in this complicated field. The methodological limitations were not very well explained. I found little support in the conclusion that the physical activity interventions were the most effective, and find it rather unlikely. This merits at least a more thorough explanation.
---

	The closing statement on "what the study adds" can, as the paper is written now, hardly be supported and needs much more nuance. The material that was retrieved merits publication however.
--	--

REVIEWER	Danai Bem University of Birmingham, Birmingham, UK
REVIEW RETURNED	01-Mar-2017

GENERAL COMMENTS	This rapid review is well written and aims to provide all the available evidence about the effectiveness of non-pharmacological interventions to improve quality of life in cancer survivors. However, the rationale of conducting a rapid review of systematic reviews and not a standard systematic review is not clearly stated in the introduction. Also, there are some important methodological issues that need to be further considered:  1. Searches were performed until June 2015; this should also be stated in the abstract. In order to be an up-to-date review the authors should re-run/update the searches. 2. i) It is not clear why the authors have chosen R-AMSTAR instead of AMSTAR (a better validated tool) for assessing the methodological quality of the included reviews. R-AMSTAR uses a quantitative approach to assess quality and doesn't allow the reviewer to weight items according to relative importance. A number of comparative studies suggest that AMSTAR is a more useful tool than R-AMSTAR. ii) Reviewers applied their own threshold for high and low quality studies. This is done in a fairly crude way where studies above a score of 33 were considered "high quality" and the rest of "low quality" irrespective of score, i.e. a study with a score of 31 was of the same "low quality" as a study that scored 16. A "high", "medium" and "low" quality grading seems more appropriate. iii) Some discussion on the most common quality issues within "low quality" studies is lacking. 3. i) Data analysis was conducted by giving greater weight to systematic reviews of higher methodological quality. However, nothing is mentioned about the quality of the included studies of those reviews. ii) Considering the limitations of quality assessment mentioned above any results should be interpreted with caution. 4. Results from Table 5 could be better illustrated in a forest plot where effectiveness trends among studies will be more visible.
---

REVIEWER	Dr Michelle M Haby Professor, Universidad de Sonora, Mexico (current full-time appointment) Honorary Senior Research Fellow, University of Melbourne, Australia (Honorary appointment)
REVIEW RETURNED	28-Apr-2017

GENERAL COMMENTS

This is a very interesting, well conducted and well written paper. Your inclusion of patients and carers in your steering group is to be congratulated and will indeed have added to the usefulness of the review. However, there are some methodological issues (as raised in my comments below) that are worth addressing to improve the transparency and accuracy of reporting.

Major comments:

1. It would be helpful for the authors to state why they are calling their paper a rapid review rather than an overview of systematic reviews (which is the term most used for systematic reviews of systematic review). What aspects of the methodology classify it as a rapid review?
2. One of the AMSTAR criteria for a high quality review includes the mention of a protocol. Also, there have been several calls for rapid reviews to be more transparent in their methods. Was there a protocol for this review? Was it published? Please include these details in the methods.
3. Another of the AMSTAR criteria is inclusion of grey literature and non-English language papers. Neither of these are included in your review. Are you able to justify this?
4. Regarding the inclusion criteria for the review I have three queries that would be worth explaining in the review:
 - 1) Why are yoga, meditation and mindfulness included when other complementary therapies are not. How are these not complementary therapies? Reference 5 is not very clear on this.
 - 2) Why is the setting limited to healthcare settings? At the same time you mention community (Table 1) – what do you mean specifically by this in regard to healthcare? Also, check your use of parenthesis in Table 1 as it is missing an).
 - 3) Your criteria for a systematic review – why doesn't it include clear inclusion criteria which is a fundamental element of a systematic review? Please provide a reference for use of these criteria if you have one.
5. Search strategy – Annex 1. Please review this strategy carefully, particularly your lack of use of parenthesis when separating two groups of OR statements with an AND statement. Should these groups of OR statements be encapsulated by parentheses? There is a spelling mistake for #3 – psychosocail* should be psychosocial*. However, if your search included the spelling mistake then you should leave it as it is.
6. Figure 1 is a little confusing – PRISMA flow diagram of study selection. It differs from the standard flow diagram for the boxes: 'studies included after title screening' and 'studies included after abstract screening'. The standard diagram has 'Records screened' and 'Full-text articles assessed for eligibility'. Also, the inclusion of a 'Records excluded' box leading from the 'Records after duplicates removed' box is confusing. It seems very strange to be able to exclude records based on title screening only and to be able to justify the reason for it without at least viewing the abstract. Can you please review this figure and include a better description of the process to aid understanding.
7. It is standard in systematic reviews to provide a list of studies excluded at full text stage (23 studies in your case). Can you please include this – even as a supplementary online file?
8. Data sources: Please clarify why you have included both PubMed and MEDLINE when these databases include basically the same content. Also, please explain why have you grouped MEDLINE and EMBASE together when they are different databases, with different

content (though there is some overlap).

9. Assessment of methodological quality. Please justify your use of AMSTAR-R instead of AMSTAR. AMSTAR includes a scoring system that gives a score from 0-11 and it is standard to classify reviews as: low quality (score of 0-3), medium quality (score of 4-7), and high quality (score of 8-11) – see for example its use in Cochrane Collaboration overviews. Further, it is the most widely used tool for assessment of methodological quality of systematic reviews.

10. Discussion, first sentence – you state that ‘only 16 reviews were included, reflecting a paucity of evidence’. How is this a ‘paucity’ of evidence? A more important consideration is the number of primary studies included in the reviews rather than the total number of reviews.

11. A PRISMA checklist has not been completed but is appropriate for reporting of rapid reviews and overviews of systematic reviews.

Minor comments:

12. Table 3: please separate the text in your columns a little more as it is difficult to read as is. Please add a list of abbreviations and check the following comments:

- a. For Buffart et al., final column includes a question (I assume to yourselves) and xx in place of the number of months.
- b. Cramer et al. and Culos-Reed et al., final column both include a question to your selves.
- c. Galveo et al. is spelt incorrectly here and at other instances in the text. A parenthesis is missing in the intervention description.
- d. Mewes et al appears to show an adverse effect in some studies, with a negative and significant effect size. Please comment on this in the text/results.
- e. Mishra et al has various missing parenthesis.
- f. Zeng et al is also missing a parenthesis in the final column.

13. Table 5:

- a. Please check Duijts et al. as there is some floating text.
- b. For Osborn et al. please add length of time to describe short and long term. There is also a floating effect size with no explanation: 1.99, 0.69 to 3.31.

14. Page 28, second sentence – please clarify if ‘between 1 and 26 weeks’ is post-exercise or post commencement of exercise.

15. Page 30, 6th line – For consistency it is best to refer to Spark’s review rather than ‘Spark’s study’ as you are not referring to an individual study.

16. Page 30, Psychological and behavioural interventions, first line – it may be worth clarifying that you are referring here to quality of the review and that this cannot be taken to infer that the individual studies included in the review were not of high quality.

17. Page 31, paragraph 2 – you end the paragraph referring to ‘physical outcomes’. This is confusing as your primary outcome is quality of life. Can you make this clearer?

VERSION 1 – AUTHOR RESPONSE

Reviewer 1

Comment:

The paper presents a rapid review of systematic reviews of non-pharmacological interventions to improve the well-being and quality of life in cancer survivors. Overall the review approach is systematic and well performed; the method of analysis however leaves some doubts that need addressing. What is meant with "rapid" in the review is not explained.

Response:

Thank you for your comments. We have removed 'rapid' and now title the study a review of reviews. Originally the review was undertaken as part of a programme development grant to inform the design of a programme grant (now successfully funded). Having spent more time on the review it is probably not appropriate to call it a rapid review.

Comment:

The authors could better format the way they present the results as there are different ways to structure quality of life and wellbeing, either through WHO based formats, EORTC or ASCO scales etc.

Response:

We have included more specific QoL measures in Table 3 but still addressing overall quality of life; Table 3 has also been updated, and more information provided.

Comment:

Further the AMSTAR score is applied in a binary way. Being the corresponding author for one of the studies (Mewes et.al) I doubt how the score was applied and whether a difference of 4 points merits the score of "Low Quality" as these are based on a low score on conflict of interest not being stated and a combined analysis which was obviously not possible due to different interventions and survivor symptoms involved. This raises doubts on the scores of the other papers and the methodological explanation is not very conclusive (page 10). Furthermore it underlines that the results should be presented in a more detailed way, as so many symptoms, phases, and interventions, (mono disciplinary, multidisciplinary, combined) can be applied. The passages on overall effectiveness suffer from the same issues and as the high/low quality label is presented consequently, this seems to "denounce" certain studies and in addition looking for aggregate scores is not very satisfying in this complicated field.

Response:

We have replaced R-AMSTAR ratings with AMSTAR ratings and have altered the ways in which 'high', 'low' and 'moderate' have been used in the body of the study. This has not made any substantial change to the overall findings.

Comment:

The methodological limitations were not very well explained.

Response:

We have included statements on the methodological limitations in the article summary on page 6 and on page 39.

Comment:

I found little support in the conclusion that the physical activity interventions were the most effective, and find it rather unlikely. This merits at least a more thorough explanation. The closing statement on "what the study adds" can, as the paper is written now, hardly be supported and needs much more nuance.

Response:

We have revised the conclusions to reflect the findings in Table 3 and 5. In place of comparing physical and psychological interventions, we have summarized what kinds of each intervention appear to be most effective.

Reviewer 2

Comment:

This rapid review is well written and aims to provide all the available evidence about the effectiveness of non-pharmacological interventions to improve quality of life in cancer survivors. However, the rationale of conducting a rapid review of systematic reviews and not a standard systematic review is not clearly stated in the introduction. Also, there are some important methodological issues that need to be further considered:

Response:

Thank you for your comments. We have changed the title to a review of reviews, and removed the term rapid, as the original intention was to inform the design of a programme. This task has been completed, and we have spent more time on the review.

Comment:

Searches were performed until June 2015; this should also be stated in the abstract. In order to be an up-to-date review the authors should re-run/update the searches.

Response:

We have updated the searches and have included new titles.

Comment:

It is not clear why the authors have chosen R-AMSTAR instead of AMSTAR (a better validated tool) for assessing the methodological quality of the included reviews. R-AMSTAR uses a quantitative approach to assess quality and doesn't allow the reviewer to weight items according to relative importance. A number of comparative studies suggest that AMSTAR is a more useful tool than R-AMSTAR.

Response:

We have used AMSTAR in place of R-AMSTAR.

Comment:

Reviewers applied their own threshold for high and low quality studies. This is done in a fairly crude way where studies above a score of 33 were considered “high quality” and the rest of “low quality” irrespective of score, i.e. a study with a score of 31 was of the same “low quality” as a study that scored 16. A “high”, “medium” and “low” quality grading seems more appropriate. Some discussion on the most common quality issues within “low quality” studies is lacking.

Response:

The AMSTAR ratings now reflect a breadth from 'low' to 'moderate' to 'high'. Low quality reviews tended not to contain a methodological assessment (AMSTAR-7, and therefore AMSTAR-8) or an assessment of publication bias (AMSTAR-10).

Comment:

Data analysis was conducted by giving greater weight to systematic reviews of higher methodological quality. However, nothing is mentioned about the quality of the included studies of those reviews. Considering the limitations of quality assessment mentioned above any results should be interpreted with caution.

Response:

We have inserted sentences on pages 6 and 39 noting this limitation.

Comment:

Results from Table 5 could be better illustrated in a forest plot where effectiveness trends among studies will be more visible.

Response:

We have considered the suggestion of a forest plot. We felt that the evidence was heterogeneous and would benefit from the individualised consideration of each study as there is little similarity in methodological approaches and bias measures. We can, if desired, do this but felt it would suggest more similarity of study designs and perhaps encourage more direct comparison across studies which may be misleading.

Reviewer 3

Comment:

This is a very interesting, well conducted and well written paper. Your inclusion of patients and carers in your steering group is to be congratulated and will indeed have added to the usefulness of the review. However, there are some methodological issues (as raised in my comments below) that are worth addressing to improve the transparency and accuracy of reporting.

Response:

Thank you for your comments.

Comment:

It would be helpful for the authors to state why they are calling their paper a rapid review rather than an overview of systematic reviews (which is the term most used for systematic reviews of systematic review). What aspects of the methodology classify it as a rapid review?

Response:

We have changed this to a review of reviews.

Comment:

One of the AMSTAR criteria for a high quality review includes the mention of a protocol. Also, there have been several calls for rapid reviews to be more transparent in their methods. Was there a protocol for this review? Was it published? Please include these details in the methods.

Response:

The original grant application included a description of what the systematic review would evaluate. We did not publish the protocol separately as we were to report within one year what the findings were for the funders, to inform a future programme grant, now successfully awarded.

Comment:

Another of the AMSTAR criteria is inclusion of grey literature and non-English language papers. Neither of these are included in your review. Are you able to justify this?

Response:

This is a study for the UK context. We concentrated on RCTs, which necessarily excluded studies that were not completed, as well as dissertations, conference papers, and others. A handful of the articles that we included did analyse grey literature and literature not in English. We have noted the limitation in the article.

Comment:

Regarding the inclusion criteria for the review I have three queries that would be worth explaining in the review: Why are yoga, meditation and mindfulness included when other complementary therapies are not. How are these not complementary therapies? Reference 5 is not very clear on this.

Response:

Yoga has been included as a form of physical exercise and perceived psychological component. Large population-based studies have shown that mindfulness is strongly correlated with greater well-being and perceived health. As stated on page 7, we followed the NHS Choices definition and exemplars of 'complementary and alternative therapies' and excluded one study on the basis of 'alternative' therapy. Most importantly, we consulted with patients in the formulation of criteria and have now stated this.

Comment:

Why is the setting limited to healthcare settings? At the same time you mention community (Table 1) what do you mean specifically by this in regard to healthcare? Also, check your use of parenthesis in Table 1 as it is missing an).

Response:

We believe that the settings are varied, from healthcare to community to home to online and believe that this is reflected in Table 3 in particular. The expression was unclear and has been amended thanks to your comment.

Comment:

Your criteria for a systematic review – why doesn't it include clear inclusion criteria which is a fundamental element of a systematic review? Please provide a reference for use of these criteria if you have one.

Response:

We believe that the PICO search strategy sets out inclusion criteria. One of these is that the systematic review had to have its own clear inclusion criteria, which we have added.

Comment:

Search strategy – Annex 1. Please review this strategy carefully, particularly your lack of use of parenthesis when separating two groups of OR statements with an AND statement. Should these groups of OR statements be encapsulated by parentheses? There is a spelling mistake for #3 – psychosocail* should be psychosocial*. However, if your search included the spelling mistake then you should leave it as it is.

Response:

Thank you for your corrections; the Annex has been amended.

Comment:

Figure 1 is a little confusing – PRISMA flow diagram of study selection. It differs from the standard flow diagram for the boxes: 'studies included after title screening' and 'studies included after abstract screening'. The standard diagram has 'Records screened' and 'Full-text articles assessed for eligibility'. Also, the inclusion of a 'Records excluded' box leading from the 'Records after duplicates removed' box is confusing. It seems very strange to be able to exclude records based on title screening only and to be able to justify the reason for it without at least viewing the abstract. Can you please review this figure and include a better description of the process to aid understanding.

Response:

We have edited the PRISMA diagram.

Comment:

It is standard in systematic reviews to provide a list of studies excluded at full text stage (23 studies in your case). Can you please include this – even as a supplementary online file?

Response:

We have included a list of 26 papers that seemed to meet criteria after the search, but which were then excluded. The reasons for exclusion are listed. This is to be a supplementary file.

Comment:

Data sources: Please clarify why you have included both PubMed and MEDLINE when these databases include basically the same content. Also, please explain why have you grouped MEDLINE and EMBASE together when they are different databases, with different content (though there is some overlap).

Response:

Although PubMed allows to search through more content than Ovid Medline, the latter allows a more focused search. Each database yields slightly different results and as we wanted to conduct a thorough search, we decided to include both. We have separated these with a comma instead of an oblique stroke (/) to mark a difference between them.

Comment:

Assessment of methodological quality. Please justify your use of AMSTAR-R instead of AMSTAR. AMSTAR includes a scoring system that gives a score from 0-11 and it is standard to classify reviews as: low quality (score of 0-3), medium quality (score of 4-7), and high quality (score of 8-11) – see for example its use in Cochrane Collaboration overviews. Further, it is the most widely used tool for assessment of methodological quality of systematic reviews.

Response:

We have now used AMSTAR in place of R-AMSTAR.

Comment:

Discussion, first sentence – you state that ‘only 16 reviews were included, reflecting a paucity of evidence’. How is this a ‘paucity’ of evidence? A more important consideration is the number of primary studies included in the reviews rather than the total number of reviews.

Response:

We have revised this in light of your comment.

Comment:

A PRISMA checklist has not been completed but is appropriate for reporting of rapid reviews and overviews of systematic reviews.

Response:

We have completed a PRISMA checklist.

Comment:

Table 3: please separate the text in your columns a little more as it is difficult to read as is. Please add a list of abbreviations and check the following comments:

Response:

We have altered the columns and added the abbreviations.

Comment:

For Buffart et al., final column includes a question (I assume to yourselves) and xx in place of the number of months.

Response:

This has been changed.

Comment:

Cramer et al. and Culos-Reed et al., final column both include a question to your selves.

Response:

This has been changed.

Comment:

Cramer et al. and Culos-Reed et al., final column both include a question to your selves.

Response:

This has been changed.

Comment:

Galveo et al. is spelt incorrectly here and at other instances in the text. A parenthesis is missing in the intervention description.

Response:

This has been changed.

Comment:

Mewes et al appears to show an adverse effect in some studies, with a negative and significant effect size. Please comment on this in the text/results.

Response:

Thank you for your comments. Reported adverse effects of interventions have been summarised under the results section on page 36.

Comment:

Mishra et al has various missing parenthesis

Response:

Thank you; we have corrected punctuation throughout the article.

Comment:

Zeng et al is also missing a parenthesis in the final column.

Response:

Thank you; we have corrected this.

Comment:

Table 5: Please check Duijts et al. as there is some floating text.

Response:

This has been addressed.

Comment:

Table 5: For Osborn et al. please add length of time to describe short and long term. There is also a floating effect size with no explanation: 1.99, 0.69 to 3.31.

Response:

Short (<8 months) versus long (>8 months) term follow up results are reported. This has now been noted in the table.

Comment:

Page 28, second sentence – please clarify if ‘between 1 and 26 weeks’ is post-exercise or post commencement of exercise.

Response:

This is post exercise intervention and has been changed.

Comment:

Page 30, 6th line – For consistency it is best to refer to Spark’s review rather than ‘Spark’s study’ as you are not referring to an individual study.

Response:

This has been changed.

Comment:

Page 30, Psychological and behavioural interventions, first line – it may be worth clarifying that you are referring here to quality of the review and that this cannot be taken to infer that the individual studies included in the review were not of high quality.

Response:

We have changed the phrasing in respect of the quality of reviews throughout given the new AMSTAR ratings. The findings have been highlighted over the question of the quality of the reviews themselves.

Comment:

Page 31, paragraph 2 – you end the paragraph referring to ‘physical outcomes’. This is confusing as your primary outcome is quality of life. Can you make this clearer?

Response:

We have changed 'physical' to 'QoL'

VERSION 2 – REVIEW

REVIEWER	W.H. van Harten Netherlands cancer Institute/University of Twente, The Netherlands no competing interest to report
REVIEW RETURNED	08-Aug-2017
GENERAL COMMENTS	The authors have done a good job in improving the paper; of course one could comment minor issues regarding one paper or the other, but that risks becoming selective. Although I agree with the scores based on the Amstar criteria as such , reviewed papers were not always written with the objective to fulfill those criteria. That leaves room for discussion on how results from well conducted studies with large accrual rates and high numbers of participants, score versus than small rct's based on lower accrual rates. I would advice spend more discussion on this issue. In all this provides a thorough overview of reviews and I now commend acceptance with a minor recommendation.

REVIEWER	Dr Michelle M Haby Profesor, Universidad de Sonora, Mexico Honorary Senior Research Fellow, University of Melbourne, Australia
REVIEW RETURNED	19-Aug-2017

GENERAL COMMENTS	. The authors have addressed my comments and I am satisfied with the changes that they have made to the manuscript. The paper is a worthy contribution to the literature. In regards to the search strategy, annex 1 - the numbers for the FULL PICO need to be updated to #1 AND #10 AND #11.
--